# Stimulation of hair follicle stem cell proliferation through an IL-1 dependent activation of γδT-cells

Pedro Lee[1†‡], Rupali Gund[2†], Abhik Dutta[2], Neha Pincha[2,3], Isha Rana[2,4], Subhasri Ghosh[2], Deborah Witherden[5], Eve Kandyba[6], Amanda MacLeod[5§], Krzysztof Kobielak[6#], Wendy L Havran[5], Colin Jamora[2*]

[1]Section of Cell and Developmental Biology, University of California, San Diego, La Jolla, United States; [2]IFOM-inStem Joint Research Laboratory, Centre for Inflammation and Tissue Homeostasis, Institute for Stem Cell Biology and Regenerative Medicine, Bangalore, India; [3]Graduate Studies, Manipal University, Manipal, India; [4]Shanmugha Arts, Science, Technology and Research Academy (SASTRA) University, Thanjavur, India; [5]Department of Immunology and Microbial Science, The Scripps Research Institute, La Jolla, United States; [6]Eli and Edythe Broad Center for Regenerative Medicine & Stem Cell Research, Department of Pathology, University of Southern California, Los Angeles, United States

*For correspondence:
colinj@instem.res.in

†These authors contributed equally to this work

Present address: ‡Department of Biochemistry and Molecular Pharmacology, NYU Langone School of Medicine, New York, United States; §Department of Dermatology, Duke University, Durham, United States; #Laboratory of Stem Cells, Tissue Development and Regeneration, University of Warsaw, Warsaw, Poland

Competing interests: The authors declare that no competing interests exist.

**Abstract** The cutaneous wound-healing program is a product of a complex interplay among diverse cell types within the skin. One fundamental process that is mediated by these reciprocal interactions is the mobilization of local stem cell pools to promote tissue regeneration and repair. Using the ablation of epidermal caspase-8 as a model of wound healing in *Mus musculus*, we analyzed the signaling components responsible for epithelial stem cell proliferation. We found that IL-1α and IL-7 secreted from keratinocytes work in tandem to expand the activated population of resident epidermal γδT-cells. A downstream effect of activated γδT-cells is the preferential proliferation of hair follicle stem cells. By contrast, IL-1α-dependent stimulation of dermal fibroblasts optimally stimulates epidermal stem cell proliferation. These findings provide new mechanistic insights into the regulation and function of epidermal cell–immune cell interactions and into how components that are classically associated with inflammation can differentially influence distinct stem cell niches within a tissue.

DOI: https://doi.org/10.7554/eLife.28875.001

## Introduction

One of the main functions of the skin is to provide the body with a barrier against external assaults while preventing excessive loss of moisture. As a result, the skin is constantly regenerating itself, but once this barrier has been breached through injury, a wound-healing response is rapidly mobilized to restore this barrier. The wound-healing response is a complex process that relies on the careful orchestration of signals coming from various cell types. Following injury, three sequential but over-lapping phases are initiated, commencing with inflammation, followed by proliferation of stem cells and concluding with tissue remodeling. Despite their temporal differences, there is significant inter-dependence among these phases that enables the restoration of tissue function (*Gurtner et al., 2008*). One such interdependency is the interaction between the inflammation and proliferation phases, which can be mediated by members of the interleukin-1 (IL-1) family of cytokines (*Dinarello, 2009*). However, the mechanism(s) by which IL-1 proteins mediate proliferation of different cell types within the repairing organ remain to be elucidated.

**eLife digest** The skin is a physical barrier that protects the body from the outside world. If the skin is injured, the body mounts a "wound healing" response to rapidly mend and restore this protective barrier. Wound healing is a complex process and relies on the different types of cells in the skin communicating with each other.

Stem cells provide tissues, like the skin, with new cells. Normally, stem cells are in a resting or inactive state. Yet, during wound healing, stem cells near the injured area are awakened and start producing more cells to repair the wound. Understanding how stem cells become activated in a wound has proved challenging because only a small number of cells near a damaged site will respond, and it is difficult to distinguish their response from that of other cells slightly further away.

Now, Lee et al. overcome this hurdle by analyzing a genetically engineered mouse in which the entire skin displays a wound healing response, even without any injury or trauma. In these mice, most of the stem cells in the skin are awakened from their normal resting state and behave as if there is a wound to heal.

It turns out that a protein called interleukin-1, which is released from damaged skin cells known as keratinocytes, can activate two different groups of stem cells in the skin to help repair the injured tissue. One group lives in the hair follicle and is normally responsible for replacing the hair that falls from the body. Lee et al. found that when the skin is wounded interleukin-1 activates certain immune cells (called γδT-cells). These immune cells then awaken the resting stem cells in the hair follicle to multiply and travel to the wound site to repair the injury. The other group of stem cells resides in the outermost layer of the skin. Interleukin-1 can also activate so-called fibroblast cells, which then stimulate this second group of stem cells to divide and cover the open wound.

Quickly healing wounds has many health benefits such as preventing infection and shortening the time to recover from an injury. These new findings may help to repair injured skin in diseases such as diabetes, where wounds can take months to heal and often leads to permanent tissue damage. The next challenge is to identify the cues that instruct the stem cells to travel to the wound site and turn into the specific cells that are required to replace the damaged cells.

DOI: https://doi.org/10.7554/eLife.28875.002

Epidermal keratinocytes are a rich source of IL-1α, which is released upon tissue damage (*Lee et al., 1997*). Unlike IL-1β, which must be proteolytically processed into its active form, IL-1α is active as a zymogen and, upon secretion from epidermal keratinocytes, can play key roles in the early inflammatory phase of the wound-healing response (*Bianchi, 2007*). In addition, IL-1α plays an important role in mediating the reciprocal crosstalk between cells within the epidermis and dermis that stimulates the secretion of stem cell proliferation factors (*Lee et al., 2009*; *Szabowski et al., 2000*). The proliferative phase of the healing program rests on the ability of various epithelial stem cell pools, such as interfollicular epidermal (IFE) stem cells and hair follicle stem cells (HfSC), to contribute progenitor cells to seal the breached epidermis rapidly (*Blanpain and Fuchs, 2009*; *Lau et al., 2009*). As a model, we used the epidermal caspase-8 knockout, which has a uniform wound-healing phenotype throughout the skin (*Lee et al., 2009*), to investigate the mechanisms by which IL-1 signaling mobilizes different epithelial stem cell pools to mount an effective wound-healing response. An understanding of the regulation of stem cell proliferation within the context of an organ will advance the goal of enhancing the regenerative process or tempering diseases that have features of a chronic wound-healing response (i.e. diseases with a 'wound signature') (*Schäfer and Werner, 2008*).

## Results

### Impact of IL-1 signaling on the proliferative phase of the wound-healing response

As we previously observed, IL-1-dependent signaling is crucial for a normal wound-healing response as mice deficient in the interleukin-1 receptor type 1 (IL1R KO) displayed a 2-day delay in wound closure (*Lee et al., 2009*; *Werner and Smola, 2001*). Interestingly, a closer analysis of the delay in

wound-closure kinetics reveals a defect that manifests itself significantly at day 3 post-wounding, suggesting a role for IL-1 signaling in the proliferative phase of the wound-healing response (*Figure 1—figure supplement 1*). This observation is consistent with previous findings that IL-1 mediates a double paracrine signaling loop wherein keratinocyte–fibroblast crosstalk generates growth factors that stimulate epidermal stem cell proliferation (*Lee et al., 2009*; *Werner and Smola, 2001*). Extending upon this concept, we observed that the interfollicular stem cell proliferation induced by a wound is significantly impaired in the absence of IL-1 signaling (*Figure 1A* and *Figure 1—figure supplement 2*). In light of these observations, we investigated whether other epithelial stem cell pools that are mobilized upon wounding were likewise dependent upon IL-1 signaling. In particular, we focused on the hair follicle stem cells (HFSCs) residing in their niche within the hair follicle known as the bulge. These bulge stem cells are responsible for the cyclical regeneration of the hair under homeostatic conditions, but are stimulated to proliferate and migrate into the epidermis following a cutaneous wound (*Ito et al., 2005*). We found that CD34+ HFSCs were highly proliferative (as noted by Ki67+/CD34+ cells) in hair follicles proximal to the wound (*Figure 1B*), but were relatively quiescent in follicles distal to the wound site (*Figure 1—figure supplement 3*). Interestingly, this increase in HFSC proliferation was also significantly dependent upon IL-1 signaling. As shown in *Figure 1B*, there was a substantial decrease in the number of CD34$^+$ HFSCs that were expressing the proliferating antigen Ki67 3 days post-wounding. Upon tracking the status of HFSC proliferation over the course of the wound-healing program, we observed that there was a quantitative decrease in proliferating HFSCs as early as one day post-wounding (*Figure 1C* and *Figure 1—figure supplement 4*). This decrease in HFSC proliferation perdured throughout the proliferative phase of the wound-healing program. Given the contribution of epithelial stem cell proliferation to the repair of the epidermis, we examined whether the impairment of both IFE stem cell and HFSC proliferation in the IL-1R KO background would manifest as a hindrance to the expansion of the epidermis upon wounding. As shown in *Figure 1D* (and *Figure 1—figure supplement 5*), the epidermal region proximal to the wound is hyperplastic, but this response is diminished in the absence of IL-1 signaling.

A major obstacle in dissecting the signaling pathways that regulate the wound response is the limited amount of cells that respond to the trauma. The heterogeneous response of cells up to 200 µm from the wound site makes it difficult to identify the extracellular cytokines and intracellular signaling pathways that orchestrate the wound-healing response in this physiological system. We addressed this constraint by employing a genetic wound-healing model based on the epidermal ablation of caspase-8 (Casp8 cKO) that converts the entire skin to a wound-like state (*Lee et al., 2009*; *Li et al., 2010*). Similar to the observations made with excisional wounds, the Casp8 cKO mouse model of wound healing displayed a thicker epidermis compared to the skin of WT mice (*Figure 1E*). Interestingly, desensitization of the cells to IL-1 signaling (via the removal of the IL-1 receptor) significantly reduced this epidermal thickening. These results confirm our previous finding that one mechanism through which epidermal stem cell proliferation is achieved is through IL-1α-mediated induction of keratinocyte proliferation factors from dermal fibroblasts (*Lee et al., 2009*). The thickened epidermal phenotype was partly due to enhanced proliferation of Keratin-5$^+$ IFE cells, which is elevated in the Casp8 cKO skin and reduced in the absence of IL-1 signaling (*Figure 1F and G*).

In order to determine whether HFSCs were also contributing to the epidermal thickening in the Casp8 cKO, the proliferation of these cells was measured. The Casp8 cKO skin had more than a two-fold increase in proliferating HFSC when compared to WT skin (*Figure 1H*). In the absence of IL-1 signaling, the number of proliferating HFSCs in the Casp8 cKO skin decreased by ~30% (*Figure 1H*). These findings were further corroborated upon detecting HFSCs with an antibody recognizing Sox9 (*Adam et al., 2015*). There was a substantial increase in Sox9 +HFSCs in the wounded skin and this increase was substantially reduced in the absence of the IL-1 receptor (*Figure 1—figure supplement 6*). Despite the significant contributions of IL-1 signaling to the development of the epidermal thickening in the Casp8 cKO line, removing IL-1 signaling was not sufficient to reduce the epidermal thickness to WT levels (*Figure 1E*). This is probably due to the contribution of other factors, such as leukocyte-derived signals that are capable of promoting stem cell proliferation, in a wound microenvironment (*Eming et al., 2007*).

An important contributor to wound-closure kinetics during the proliferative phase of wound healing is cell migration. Moreover, it has previously been reported that IL-1α can stimulate epidermal keratinocyte migration (*Chen et al., 1995*). We therefore explored whether IL-1α can similarly serve

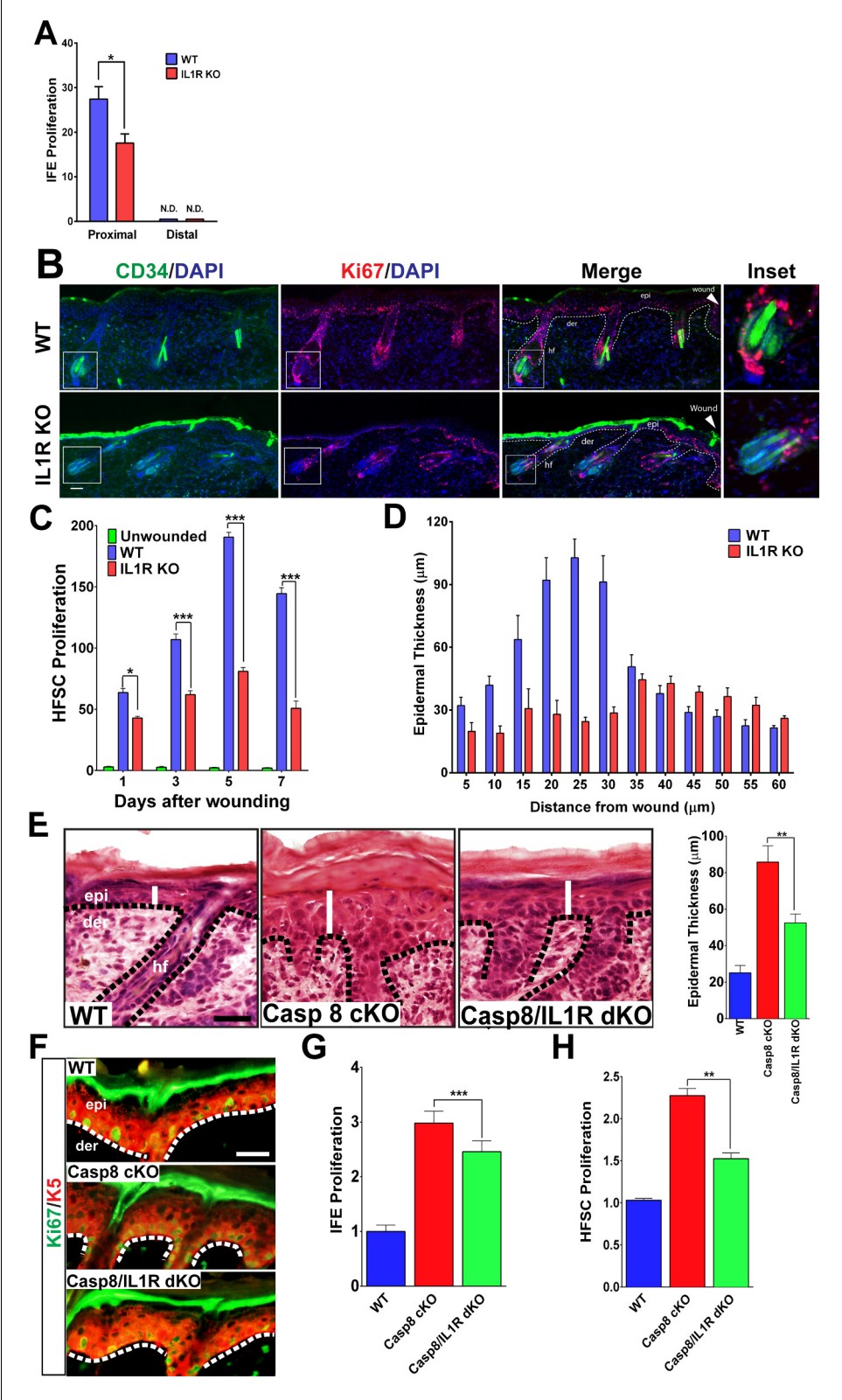

**Figure 1.** Impact of IL-1 signaling on epidermal stem cell proliferation. (**A**) Quantification of proliferating cells in the interfollicular epidermis (IFE) of the skin from WT or IL1R-KO 8-week-old mice 3 days post-wound. The proximal area of measurement was restricted to the first three hair follicles adjacent to the wound edge. The cell count from a region 2 cm away from the wound centre was considered to be distal. (**B**) Immunostaining of day 3 wound proximal skin sections from 8-week-old WT or IL-1R KO mice for CD34 (green) and Ki67 (red) to mark proliferating hair follicle stem cells (HFSCs). Nuclei

*Figure 1 continued on next page*

*Figure 1 continued*

were stained with DAPI (blue). Scale bar = 50 µm. The dotted lines represent the epidermal–dermal interface. The inset shows the magnified view of a hair follicle with proliferating cells near the wound. (C) Quantification of proliferating HFSCs present near the wound in WT or IL-1R-deficient mice compared to unwounded mice. (D) The epidermal thickness was quantified by measuring the distance between the keratin-5+ basal layer and the loricrin+ granular layer, starting from the region with keratin-5 expression in the wound. Each bar in the graph is a mean of three data points obtained from 50-µm-wide windows. (E) H&E staining of postnatal day 4 (P4) skin showing epidermal thickness (denoted by white lines) between the various genotypes, WT = wildtype, C8CKO = caspasecKO and C8/IL1R = caspase-8/IL1R dKO. The dotted black lines represent the basement membrane separating the epidermis (epi) and the dermis (der). hf denotes hair follicle. Scale bar = 50 µm. Quantitation of epidermal thickness is represented in the histogram as the average ±SEM. (F) Hyperproliferation of P4 epidermis in the KO skin is revealed by increased expression of Ki67 (green) and is reduced in the caspase-8/IL1R dKO. K5 = keratin 5 (red). (G) Quantitation of proliferating interfollicular epidermal cells is shown as the average ± SEM relative to WT levels. (H) Proliferation of hair follicle stem cells. The data shown are the fold difference in different genotypes relative to WT levels that are set to 1. The data shown in E–H are from six different mice per genotype. **p<0.001, ***p<0.0001.

DOI: https://doi.org/10.7554/eLife.28875.003

The following figure supplements are available for figure 1:

**Figure supplement 1.** Wound closure rate in the skin of WT and IL1R KO mice.
DOI: https://doi.org/10.7554/eLife.28875.004
**Figure supplement 2.** Representative image for *Figure 1A*.
DOI: https://doi.org/10.7554/eLife.28875.005
**Figure supplement 3.** Proliferation of HFSCs in a distal wound region.
DOI: https://doi.org/10.7554/eLife.28875.006
**Figure supplement 4.** Representative image of *Figure 1C*.
DOI: https://doi.org/10.7554/eLife.28875.007
**Figure supplement 5.** Representative image of *Figure 1D*.
DOI: https://doi.org/10.7554/eLife.28875.008
**Figure supplement 6.** Sox9 expression in wounded skin.
DOI: https://doi.org/10.7554/eLife.28875.009
**Figure supplement 7.** In vitro migration of HFSCs.
DOI: https://doi.org/10.7554/eLife.28875.010

as a chemoattractant to stimulate HFSC emigration from the bulge to the wound site. To separate the effect of IL-1α on HFSC migration from proliferation unambiguously, we reconstituted the migration in a transwell assay (*Figure 1—figure supplement 7*). When treated with conditioned media (CM) from the Casp8 cKO epidermis, primary mouse HFSCs showed a >3 fold increase in chemotaxis relative to cells exposed to CM from wild-type epidermis. Interestingly, HFSCs lacking the IL-1R remained responsive to the chemotactic cue in the Casp8 cKO CM. This finding is consistent with the observation that recombinant IL-1 was incapable of inducing the transwell migration of HFSCs. Altogether, this provides evidence that IL-1α is neither necessary nor sufficient to induce HFSC migration.

## Loss of γδT-cells phenocopies the deficiency of IL-1 signaling in the epithelial compartment of the wound-like skin

As IL-1 signaling is classically associated with the establishment of localized inflammation, we interrogated the inflammatory response in the wound lacking the IL-1R. Among the earliest inflammatory cells that respond to injury are macrophages and granulocytes, which are known to enhance the proliferation of epidermal stem cells (*Martin, 1997*). In the Casp8 cKO skin, the numbers of macrophages and granulocytes were increased and a marked reduction in the numbers of these cells was observed in the absence of IL-1 signaling (*Figure 2A*). Similarly, other well-established contributors to the wound-healing program are epidermal resident γδT-cells, also known as dendritic epidermal T-cells (DETCs). Following a stress signal such as wounding, γδT-cells become activated and promote epidermal stem cell proliferation by secreting keratinocyte growth factor (KGF, also known as FGF7) (*Jameson and Havran, 2007*). Wound healing is impaired in γδT-cell deficient mice, which underscores the important role of these immune cells in the repair process (*Jameson et al., 2002*). Furthermore, γδT-cells play an important role in the recruitment of other immune cells such as macrophages during wound healing (*Jameson et al., 2005*). In the Casp8 cKO/Tcrd$^{-/-}$ (γδT-cell KO) mouse, the number of granulocytes and macrophages were decreased (*Figure 2A*). This impairment

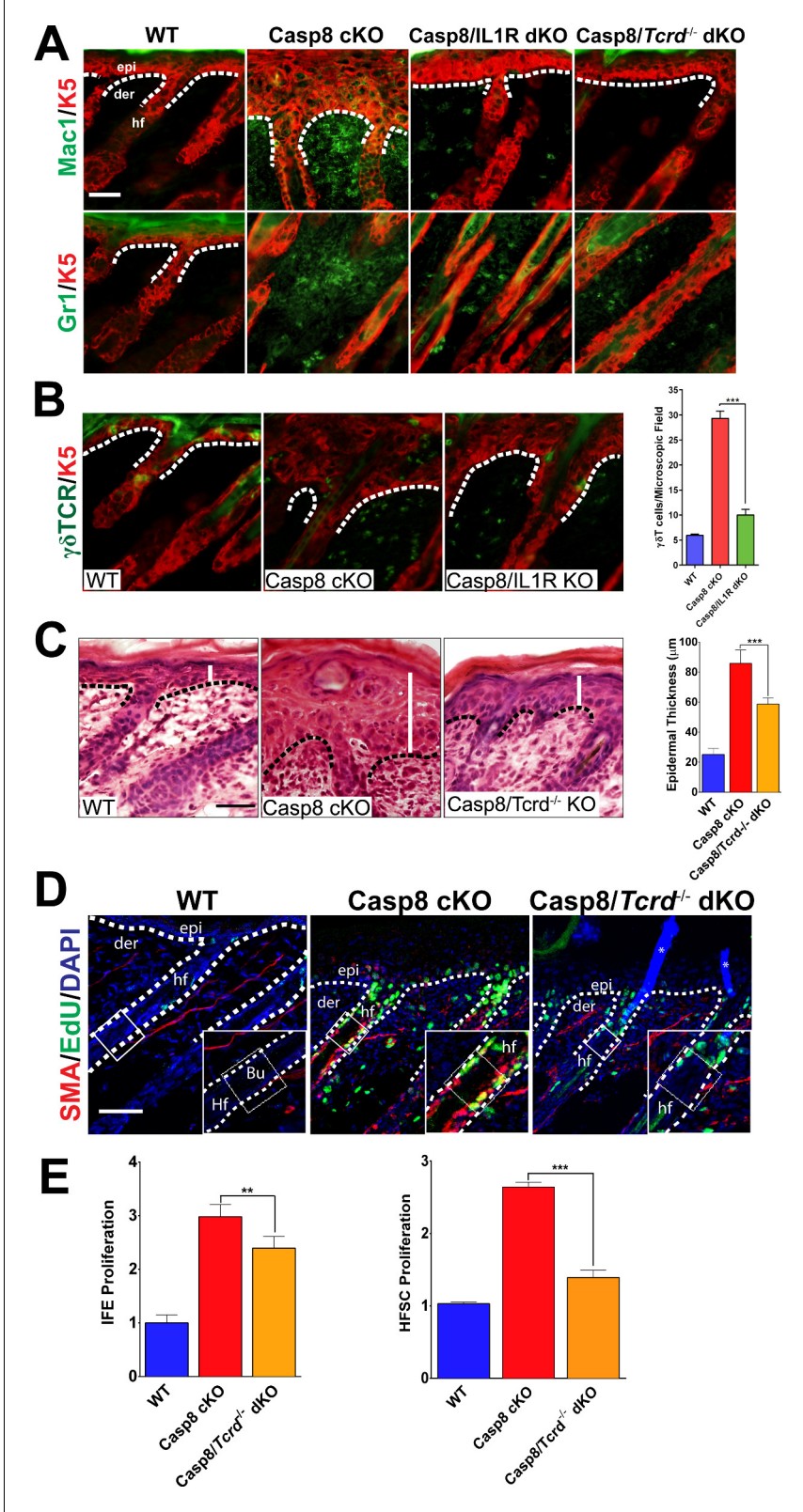

**Figure 2.** Loss of γδT-cells phenocopies the deficiency of IL-1 signaling in C8- KO epidermis. (**A**) Immunofluorescence of mouse skin sections from different genotypes. Staining for keratin 5 (K5, red) and macrophages (Mac1, green) or for granulocytes (Gr-1, green) reveals a reduction in innate immune cell infiltration in the caspase-8 conditional knockout (C8 KO) skin in the absence of IL-1 signaling (C8/IL1R dKO) or γδT-cells (C8/γδTCR dKO). Scale bar = 50 μm, the white dotted lines denote basement membrane. (**B**) Immunofluorescence of γδT-cells (green) and keratin 5 (K5, red) in the skin

*Figure 2 continued on next page*

*Figure 2 continued*

of mice from different genotypes identified in the figure. The histogram shows that there is an increase in the amount of γδT-cells in the C8KO skin relative to the wild-type control, and this is reduced in the absence of IL-1 signaling (C8/IL1R dKO). (C) H&E staining of P4 skin showing epidermal thickness differences denoted by the white lines. The dotted line denotes the basement membrane separating epidermis (epi) and dermis (der), hf denotes a hair follicle. Scale bar = 50 µm. The quantified data from three mice per genotype are shown in the histogram. (D) Detection of proliferating cells by EdU labeling (green). The bulge region where HFSCs reside (shown as a box along the hair follicle) was marked by smooth muscle actin staining (red) of the arrectorpilli muscle. Nuclei were stained by DAPI (blue). The dotted line represents the epidermal–dermal interface. The asterisk marks the non-specific immunofluorescence from DAPI channel; the inset shows the magnified view of the hair bulge. (E) Quantification of the proliferating interfollicular epidermal cells (IFE) and hair follicle stem cells (HFSC) from (D). Data shown in B, C, and E are from six different mice per genotype. **p<0.001, ***p<0.0001.

DOI: https://doi.org/10.7554/eLife.28875.011

The following figure supplements are available for figure 2:

**Figure supplement 1.** Loss of γδT-cells phenocopies the proliferation deficiency of IL-1 signaling in excisional cutaneous wounds.
DOI: https://doi.org/10.7554/eLife.28875.012

**Figure supplement 2.** Effect of γδT-cells on IFE proliferation.
DOI: https://doi.org/10.7554/eLife.28875.013

**Figure supplement 3.** Effect of γδT-cells on HFSC proliferation.
DOI: https://doi.org/10.7554/eLife.28875.014

**Figure supplement 4.** Effect of γδT-cells on epidermal thickness.
DOI: https://doi.org/10.7554/eLife.28875.015

**Figure supplement 5.** Extracellular IL-1α in the genetic wound-healing model.
DOI: https://doi.org/10.7554/eLife.28875.016

**Figure supplement 6.** Quantification of secreted IL-1α from excisional wounds.
DOI: https://doi.org/10.7554/eLife.28875.017

**Figure supplement 7.** Representative image of day one post-wound in *Figure 2—figure supplement 6*.
DOI: https://doi.org/10.7554/eLife.28875.018

of immune cell recruitment is similar to the effect observed upon removal of IL-1 signaling in the Casp8 cKO (*Figure 2A*).

On the basis of their shared effect on immune cell recruitment, we investigated whether there is an epistatic relationship between IL-1 signaling and γδT-cell activity. Interestingly, γδT-cell numbers are increased in the Casp8 cKO mice (*Lee et al., 2009*), but in the absence of IL-1 signaling, there is a dramatic reduction of the numbers of these cells in the skin (*Figure 2B*). Given the ability of γδT-cells to stimulate epidermal stem cell proliferation, we hypothesized that the removal of these cells would reduce the epidermal hyperplasia in the Casp8 cKO mouse as does the loss of IL-1 signaling. We found that the epidermal thickness in the Casp8 cKO/Tcrd$^{-/-}$ mouse was significantly reduced in comparison to that in the Casp8 cKO mouse (*Figure 2C*).

We then assessed the status of HFSCs in the Casp8 cKO background and its dependence on the activity of γδT-cells. Proliferating HFSCs were marked by a pulse of EdU, which is incorporated into the replicating DNA of dividing cells. Owing to antibody incompatibility, we visualized the arrector pili muscle (APM) with α-smooth muscle actin as a landmark for the HFSC niche (the bulge), which resides at the terminus of the APM (*Poblet et al., 2002*). In the wild-type skin, there is a minimal amount of EdU, whereas the Casp8 cKO skin exhibits robust EdU signal in the bulge region (*Figure 2D*). However, this elevated proliferation is markedly reduced in the Casp8 cKO skin lacking functional γδT-cells. Interestingly, the impact of γδT-cell activity is more pronounced upon HFSC proliferation than in IFE cells in the Casp8 cKO skin (*Figure 2E*). This same reliance on γδT-cells activity for optimized stem cell proliferation was also observed when the stimulus was an excisional wound (*Figure 2—figure supplements 1–3*). Likewise, the impairment of epithelial stem cell proliferation in the skin resulting from compromised γδT-cells drastically reduces the epidermal hyperplasia that accompanies the wound-healing program (*Figure 2—figure supplement 4*).

We also tested the possibility that IL-1 signaling exerts its effect separately from γδT-cells. We thus examined whether the expression of IL-1α is affected in the absence of γδT-cells. As shown in *Figure 2—figure supplement 5*, the amount of extracellular IL-1α was comparable in Casp8 cKO skin and Casp8 cKO/Tcrd$^{-/-}$ skin. This was verified by quantitation of IL-1 secretion in wounded samples from wild-type and Tcrd$^{-/-}$ skin, wherein the IL-1α levels were not significantly different (*Figure 2—figure supplements 6–7*). These data point to the fact that, despite normal levels of IL-

1α in the wound, active γδT-cells are necessary to elicit the early stimulation of HFSC proliferation. Altogether, these observations suggest that IL-1 signaling converges upon γδT-cells to mediate epithelial stem cell proliferation in the skin.

## Activation of γδT-cells by IL-1

As IL-1 signaling elicits many of the same effects as activated γδT-cells, we then focused on interrogating whether there is a direct relationship between IL-1 signaling and γδT-cell function. The remarkable ability of γδT-cells to perform a variety of effector functions depends on their ability to recognize stress signals coming from keratinocytes (*Jameson and Havran, 2007*). Unlike αβ T-cells, which recognize protein fragments processed by antigen presenting cells, the effector function of γδT-cells depends on a combination of a yet unknown antigen recognized by the γδ-TCR and a variety of co-stimulatory molecules (*Witherden et al., 2010*). In particular, keratinocytes that are stressed during wound healing increase the expression of the coxsakie and adenovirus receptor (CAR), which acts as a co-stimulatory ligand that is recognized by the junctional adhesion molecule-like protein (JAML) on the surface of γδT-cells (*Witherden et al., 2010*, *Witherden and Havran, 2011*). Upon activation by a wound, γδT-cells undergo a variety of changes that are hallmarks of an activated cell, including the loss of dendritic extensions, increased proliferation and secretion of cytokines and growth factors that are essential for wound healing (*Jameson and Havran, 2007*). Among the functional consequences of γδT-cell activation is enhanced proliferation of epidermal stem cells (*Jameson et al., 2002*) and, as shown in *Figure 2E*, a stronger effect on HFSC proliferation. As epithelial stem cell proliferation was decreased in the caspase-8/IL1R dKO (*Figure 1H*), we hypothesized that IL-1 signaling is required for optimal activation of γδT-cell, which, in turn, promotes the proliferation of cutaneous epithelial stem cells.

Using the Casp8 cKO and caspase-8/IL1R dKO mice, we investigated whether a parameter of γδT-cell activation, namely proliferation, is dependent upon IL-1 signaling. We assayed γδT-cell proliferation and found that the number of proliferating γδT-cells was increased three-fold in the Casp8 cKO skin compared to WT controls, whereas a significant decrease was observed in the absence of IL-1 signaling (*Figure 3A*). Similarly, in excisional wounds, elevated γδT-cell proliferation is dependent on IL-1 signaling (*Figure 3—figure supplements 1* and *2*). It is noteworthy, however, that γδT-cells in the caspase-8/IL1R dKO still displayed increased proliferation compared to wild-type cells, which may reflect the redundancy in this system. We also observed increased γδT-cell activation in the caspase-8/IL1R dKO mice, as observed by TNFα expression (*Figure 3B and C*). The frequency of these activated γδT-cells was markedly reduced in the absence of IL-1 signaling. In the newborn thymus, IL-1 signaling has been shown to have a synergistic effect with IL-7 in enhancing the proliferation of γδT-cells (*Lynch and Shevach, 1992*). Epidermal keratinocytes constitutively express IL-7 and this expression is required for γδT-cell development (*Jameson and Havran, 2007*). In the Casp8 cKO, IL-7 expression is upregulated and this is not affected by the absence of IL-1 signaling (*Figure 3—figure supplement 3*). Similarly, IL-7 expression is not affected in Casp8 cKO skin or excisional wounds that lack γδT-cells (*Figure 3—figure supplements 4* and *5*). On the basis of this observation, we hypothesized that IL-1α released from the keratinocytes in the Casp8 cKO mouse can act synergistically with IL-7 to enhance γδT-cell proliferation.

In order to validate the contributions of IL-1 and IL-7 signaling to γδT-cell proliferation, we established short-term γδT-cell in vitro cultures from WT mice (*Witherden et al., 2010*). The use of this in vitro approach allowed us to reconstitute the different components required to stimulate γδT-cell proliferation. We collected conditioned media (CM) from WT and Casp8 cKO keratinocytes (*Lee et al., 2009*), applied them to WT or IL1R$^{-/-}$ γδT-cells and monitored the proliferation rates of these cells. γδT-cells were cultured in activating conditions using anti-CD3 and anti-JAML as previously described (*Witherden et al., 2010*), and then incubated for 2 days in the presence of CM, recombinant human IL-1α (rhIL-1α), recombinant human IL-7 (rhIL-7), or an IL-7 inhibitory antibody. The number of γδT-cells increased 2.5-fold when they were treated with Casp8 cKO CM rather than WT CM (*Figure 3D*). Conversely, removing IL-1 signaling by treating IL1R$^{-/-}$ γδ T-cells with Casp8 cKO CM resulted in significantly reduced proliferation. Similarly, blocking IL-7 signaling in the Casp8 cKO CM with an anti-IL-7 inhibitory antibody blocked γδT-cell proliferation. This effect was enhanced when both IL-1 and IL-7 signaling were perturbed in tandem. As epidermal keratinocytes constitutively secrete IL-7, we predicted that adding recombinant human IL-1α (rhIL1α) to WT CM would be sufficient to cause an increase in γδT-cell proliferation. Consistent with this hypothesis, the addition

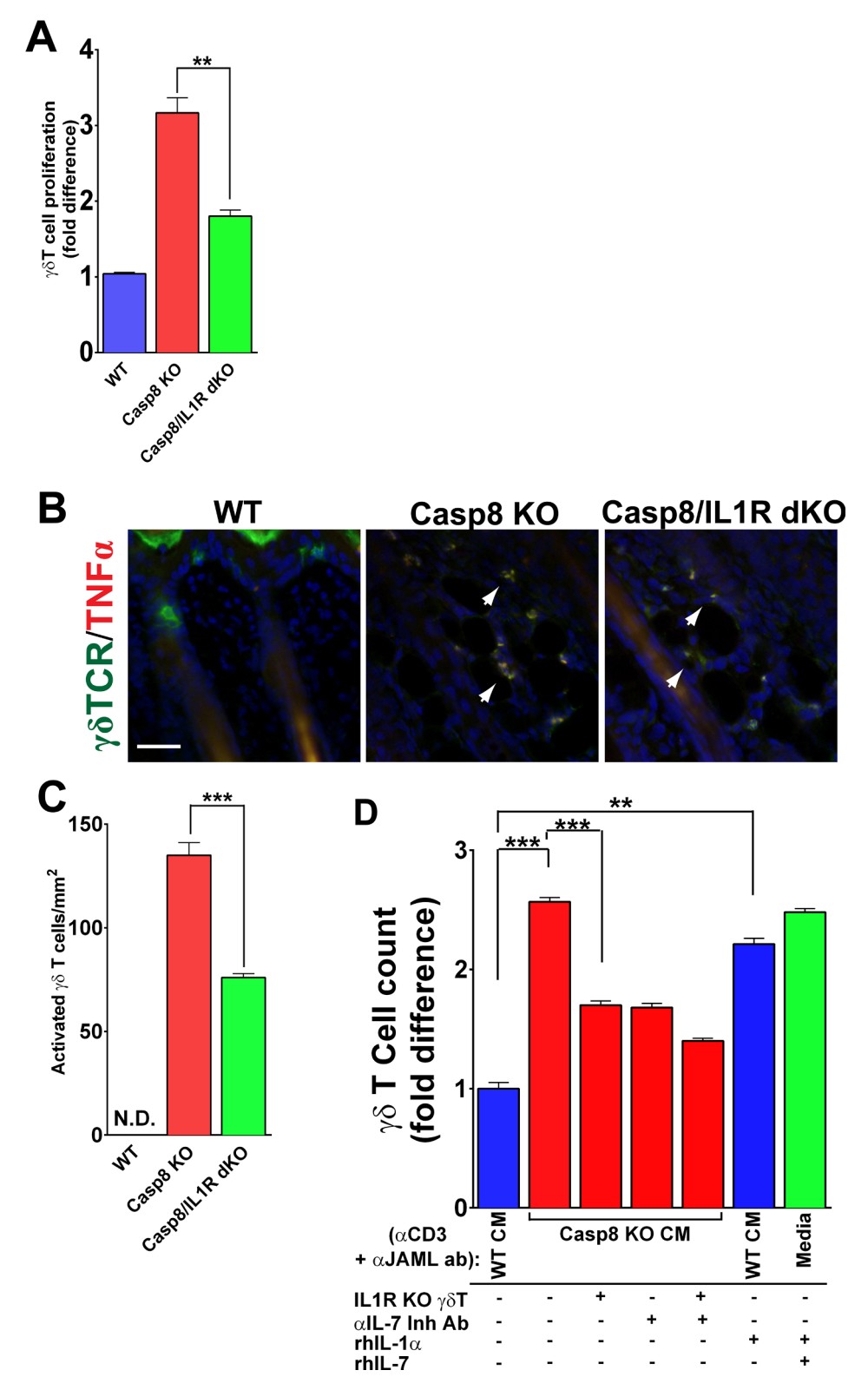

**Figure 3.** IL-1 signaling contributes to γδT-cell proliferation in the caspase-8 cKO skin. (A) Quantification of γδT-cell proliferation. The numbers of γδT-cells are reported as the fold difference from WT, which was normalized to 1. **p<0.001. (B) Detection of activated γδT-cells. γδT-cell (green) activation was monitored by expression of TNFα (red) via immunofluorescence. Nuclei (blue) are stained with DAPI. Arrowheads mark some of the activated γδT-cells. (C) Quantification of activated γδT-cells per square millimeter. Scale bar = 50 μm, ***p<0.0001. (D) Recapitulation of γδT-cell proliferation in vitro.

*Figure 3 continued on next page*

Figure 3 continued

Primary γδT-cells were isolated from the skin of either wild type (WT) or IL1R knockout mice (IL1R$^{-/-}$ γδT-cell), primed with anti-CD3 and anti-JAML and treated with conditioned media from WT (WT CM) or caspase-8 conditional knockout (C8 KO CM) skins. Inhibitory antibody against IL-7 or recombinant human IL-1α or IL-7 were added as indicated. The histogram shows the fold difference in γδT-cell numbers relative to numbers of WT CM-treated γδT-cells, which are normalized to 1. Data are representative of at least three independent experiments performed in triplicate. **p<0.001, ***p<0.0001.

DOI: https://doi.org/10.7554/eLife.28875.019

The following figure supplements are available for figure 3:

**Figure supplement 1.** Proliferation of γδT-cells post wounding.
DOI: https://doi.org/10.7554/eLife.28875.020

**Figure supplement 2.** Quantification of proliferating γδT-cells.
DOI: https://doi.org/10.7554/eLife.28875.021

**Figure supplement 3.** IL7 expression from caspase 8-deficient epidermis is not affected by IL-1α qPCR measurements of levels of IL-7 expression by keratinocytes.
DOI: https://doi.org/10.7554/eLife.28875.022

**Figure supplement 4.** IL-7 expression in excisional wounds.
DOI: https://doi.org/10.7554/eLife.28875.023

**Figure supplement 5.** IL-7 expression in a mouse model of wound healing as revealed by qPCR measurements of levels of IL-7 expression by keratinocytes.
DOI: https://doi.org/10.7554/eLife.28875.024

of rhIL-1α to WT CM elicited an increase in γδT-cell proliferation comparable to the effect seen with Casp8 cKO CM. In fact, simply adding recombinant IL-1 and IL-7 to normal growth media is sufficient to reconstitute the γδT-cell proliferation observed in the Casp8 cKO CM (*Figure 3D*). Altogether, these results support the model in which IL-1 and IL-7 signaling, in conjunction with γδT-cell activation, amplifies the proliferation of these cells in response to stresses that alter epidermal homeostasis.

## Contributions of keratinocyte–fibroblast and keratinocyte–γδT-cell interactions to stem cell proliferation

We have demonstrated that stem cells residing in distinct niches of the skin (i.e. the interfollicular epidemis and the hair follicle bulge region) are modulated by IL-1 signaling in a wound-like environment. The data presented to date and reported previously (*Lee et al., 2009*) suggest that the effect of IL-1 signaling on different stem cell pools is mediated through the dichotomous activation of dermal fibroblasts and γδT-cells. However, in the skin, where multiple cellular interactions are occurring simultaneously in an intricate manner, it is difficult to resolve the direct effects of IL-1 signaling on each subset of stem cells. As IL-1 signaling can stimulate different pools of epithelial stem cells in the skin, we sought to interrogate whether activated fibroblasts and γδT-cells can induce proliferation within distinct epithelial stem cell niches. Owing to a lack of definitive markers to distinguish IFE cells from HFSCs, assaying the proliferation of these cells within the skin is a challenge. On the other hand, in vitro cultures of IFE cells or HFSC are well established, and grafting experiments have shown that these cells maintain their progenitor properties even after several passages in vitro (*Blanpain et al., 2004*; *Ghazizadeh and Taichman, 2005*). In order to delineate the contributions of keratinocytes–fibroblast interactions to epithelial stem cell proliferation, we activated dermal fibroblasts (df) with Casp8 cKO CM or rhIL-1α, as evidenced by their expression of FGF7 and GM-CSF (*Figure 4A*). CM was collected from these activated df and applied to IFE cells or HFSC to test their effect on the proliferation of these two different epithelial stem cells. Treatment with CM from activated df caused a dramatic increase in the proliferation of IFE cells and a modest increase in HFSC proliferation (*Figure 4B*). In the absence of IL-1 signaling, however, the proliferation of both IFE cells and HFSCs was reduced to that of control levels (*Figure 4B*, *Figure 4—figure supplement 1*). These observations suggest that IFE cells preferentially respond to signals from mesenchymal cells.

The treatment of primary γδT-cell cultures with Casp8 cKO CM resulted in their activation, as demonstrated by their increased expression of FGF7 and TNFα (*Figure 4C*). Moreover, abrogating IL-1 or IL-7 signaling was able to reduce FGF7 and TNFα expression, whereas rhIL-1α or rhIL-7 was sufficient to induce maximal FGF7 expression. Exposure of HFSCs to CM from activated γδT-cells caused a significant increase in the proliferation of the HfSC (*Figure 4D*). Although activated γδT-

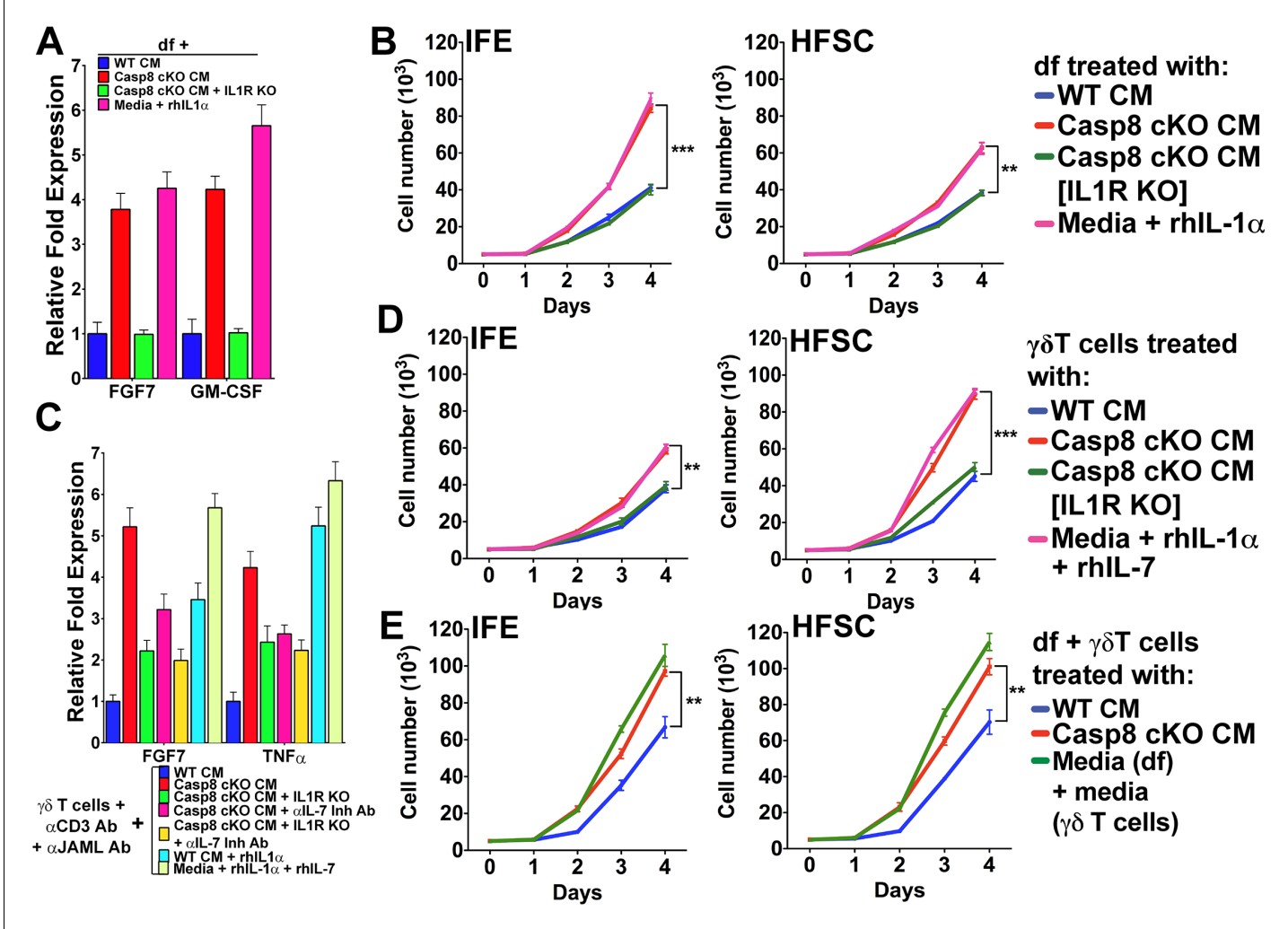

**Figure 4.** Conditioned media from dermal fibroblasts and activated γδT-cells differentially enhance epithelial stem cell proliferation in distinct niches within the skin. (A) Activation of dermal fibroblasts (df). df were treated with conditioned media (CM) as noted in the figure and the expression of the growth factors FGF7 and GM-CSF were assessed by qPCR. (B) Effect of activated df on stem cell proliferation. The proliferation rates of isolated interfollicular epidermal (IFE) and hair follicle stem cells (HFSC) were determined by cell counting with trypan blue exclusion following treatment with the CM from dermal fibroblasts. Dermal fibroblasts were first activated as described in the figure legend prior to conditioning fresh medium with them. Cell numbers were determined daily over four days. (C) Potentiation of γδT-cell activation by CM. WT or IL1R$^{-/-}$ γδT-cells were treated with CM as described in the figure, and the markers for γδT-cell activation, FGF7 and TNFα, were assessed by qPCR. (D) Effect of activated γδT-cells on stem cell proliferation. The proliferation rates of isolated IFE and HFSC were determined by trypan blue exclusion cell counting. IFE and HFSC were incubated for four days with CM from γδT-cells. (E) Combinatorial effect of activated df and γδT-cells on stem cell proliferation. The proliferation rates of isolated IFE and HFSC were determined by trypan blue exclusion cell counting. IFE and HFSC were incubated for four days with CM from df and γδT-cells that were first treated as described in the figure. Data in (A) and (C) were obtained from triplicates of at least three independent experiments and are represented as the fold difference ± SEM. The data shown in (B), (D) and (E) are the averages ± SEM of three independent experiments analyzed in triplicate. **p<0.001, ***p<0.0001.

DOI: https://doi.org/10.7554/eLife.28875.025

The following figure supplements are available for figure 4:

**Figure supplement 1.** IFE and HFSC proliferation invoked by activated fibroblasts and γδT-cell-conditioned media.
DOI: https://doi.org/10.7554/eLife.28875.026

**Figure supplement 2.** Expression of other IL-1 family members in the dermis.
DOI: https://doi.org/10.7554/eLife.28875.027

**Figure supplement 3.** Expression of IL-1 family members by activated γδT-cells.
DOI: https://doi.org/10.7554/eLife.28875.028

cells can also stimulate IFE cell proliferation, the numbers of these cells were 40% smaller than those of HFSCs. Similarly, CM from rhIL-1α- or rhIL-7-treated γδT-cells was sufficient to recapitulate the increased proliferation of the HFSCs (*Figure 4D*).

As the CM from df and γδT-cells can enhance the proliferation of the different stem cell pools, we sought to investigate whether the effects on epithelial stem cell proliferation would be additive. Indeed, treatment of both IFE cells and HFSCs with a combination of CM from activated df and γδT-cells substantially increased their proliferation rates (*Figure 4E*). One probable reason for the differential proliferation of the IFE cells and HFSCs is the difference in components present in the cytokine cocktail secreted from either df or γδT-cells. For instance, even though both activated df and γδT-cells can secrete the mitogenic factor FGF7, only γδT-cells secrete IL1F8 or IL-36β (*Figure 4—figure supplements 2* and *3*), which were recently shown to cause proliferation of stem cells (*Yang et al., 2010*). IL1F8 signals through the IL-36 receptor and it is possible that HFSCs are more susceptible to the mitogenic effects of this cytokine. On the basis of these observations, we conclude that IL-1-mediated activation of γδT-cells preferentially promotes HFSC proliferation.

## Discussion

One of the goals of regenerative medicine is to restore the functional state of the tissue swiftly following trauma and/or disease. A major obstacle to manipulating this process in mammalian tissues is the large number of cells that carry out a synchronized effort in the regenerative/repair process. Therefore, understanding the coordinated interactions between the multiple cell types that orchestrate a successful repair program is a high priority. The efficient repair of damaged tissue depends upon the interactions between the inflammatory, proliferative and remodeling phases of the wound-healing response (*Gurtner et al., 2008*). In this study, we employed a genetic mouse model of wound healing, namely the conditional knockout of epidermal caspase-8. Our previous work has demonstrated that the downregulation of epidermal caspase-8 is a normal phenomenon upon wounding of the mouse skin (*Lee et al., 2009*). Moreover, we demonstrated that the conditional ablation of epidermal caspase-8 was able to recapitulate many of the cardinal features of a cutaneous wound-healing response. This genetic mouse model is particularly useful in efforts to unravel the signaling network of the wound-healing program, given that the entire skin essentially behaves as a wound. By contrast, cellular responses to an excisional wound to the skin are limited to the cells immediately proximal to the trauma. As a consequence, there is poor signal to noise ratio given the low number of cells that are responding to the wound surrounded by a large number of cells that are in their normal homeostatic mode. This is particularly relevant to studies of hair follicle stem cell activation during wound healing. Only a handful of hair follicles exhibit activated stem cells upon trauma to the skin, whereas the caspase-8 conditional knockout animals exhibit stem cell activation in the vast majority of the hair follicles within the knockout skin.

The focus of this study was to understand how one of these signaling nodes that is an important mediator of inflammation (IL-1 signaling) manages to enhance the proliferation of different stem-cell pools within the skin. IL1α is released immediately upon wounding, downstream of reduction of caspase-8 expression (*Lee et al., 2009*), and mediates the activation of dermal fibroblasts which then stimulate epidermal stem cells (*Figure 5*). In the present study, we demonstrate that IL-1 signaling can partner with IL-7 (which is constitutively secreted from epidermal keratinocytes) to expand the population of active γδT-cells. These activated γδT-cells can then stimulate the proliferation of stem cells within the bulge of the hair follicle and can mobilize them for epidermal wound repair. As the wound-healing response progresses, given the redundancy of the system, it is likely that other infiltrating cells can contribute to (or take over the duties of) stimulating epithelial stem cell proliferation. One illustrative case of this redundancy is in the activation of HFSCs by multiple routes. We hypothesize that the IL-1 +IL-7/γδTcell/HFSC proliferation module described herein is the rapid mechanism that stimulates HFSC proliferation using cells already residing in the tissue. Interestingly, wounded keratinocytes can also produce CCL2 to attract macrophages to a wound (*Osaka et al., 2007*). These activated macrophages in turn secrete IL-1β (a more potent relative of IL-1α) to induce hair regrowth, and it is tempting to postulate that this mechanism maintains and potentiates the effects of IL-1 +IL-7/γδTcells on HFSC proliferation.

IL-1 signaling has been studied in depth and found to play critical roles in a variety of cellular processes from immune responses to the proliferation of epidermal stem cells (*Dinarello, 2009*).

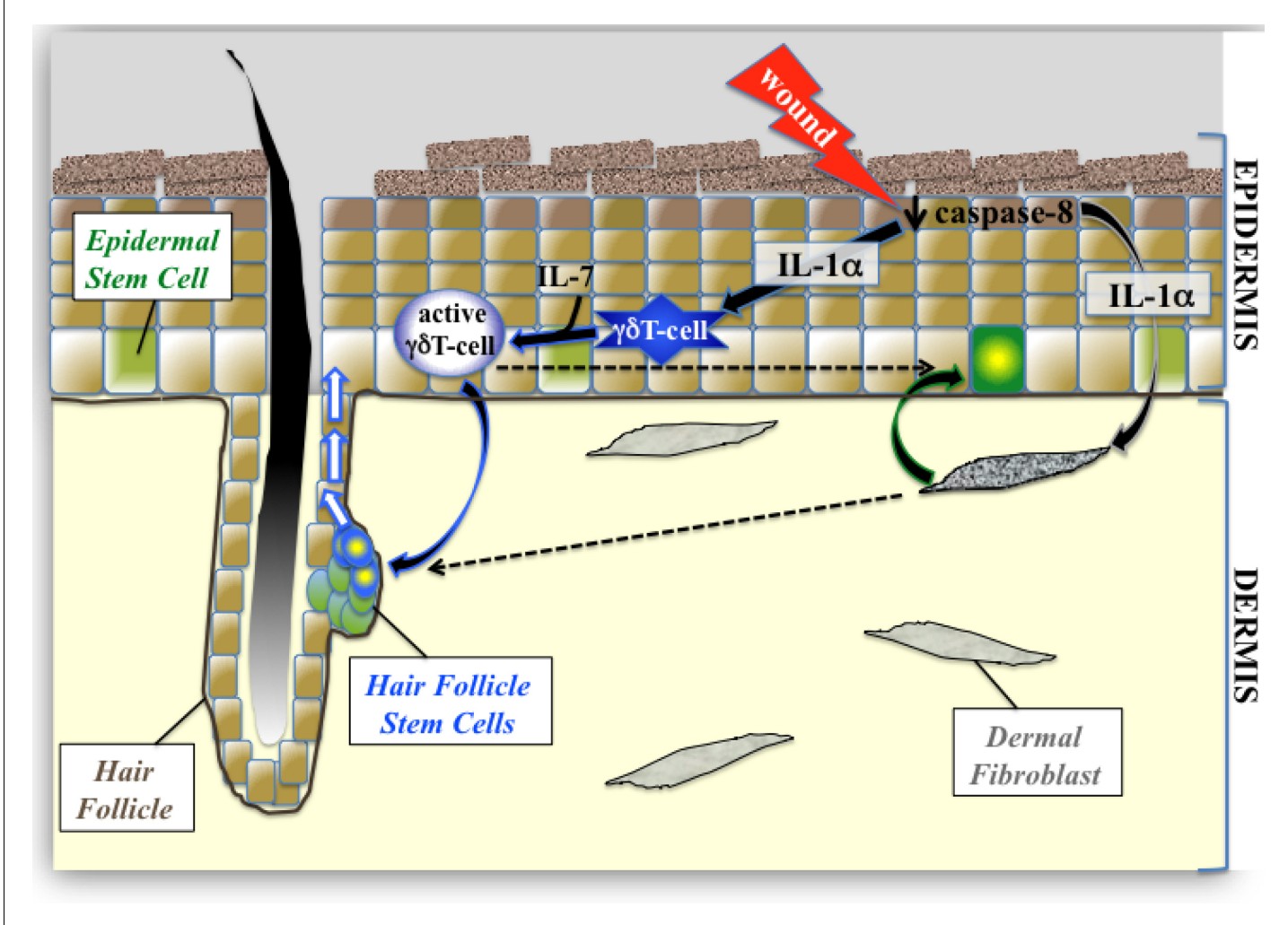

**Figure 5.** Proposed model of epidermal stem cell proliferation in the caspase-8 conditional knockout model of wound healing. Upon wounding, caspase 8 levels are downregulated causing the release of IL-1α, which can stimulate the proliferation of interfollicular epidermal stem cells (IFE) through its interactions with dermal fibroblasts. This IL-1αmediated epithelial–mesenchymal crosstalk has a milder effect on hair follicle stem cell (HFSC) proliferation. IL-1α and IL-7 cooperate to enhance the proliferation of activated γδT-cells, which preferentially influences HFSC proliferation.
DOI: https://doi.org/10.7554/eLife.28875.029

Despite the pleiotropic effects of IL-1 signaling in various cell types and its important function during wound healing, blocking IL-1Receptor-1-dependent signaling does not completely impair the repair process. This can be attributed in part to compensatory mechanisms involving other IL-1 family members. IL-1 family members such as IL-1F6, IL-1F8 and IL-1F9 are largely found in epithelial tissues. Moreover, transgenic mice overexpressing IL-1F6 in the basal layer of the epidermis develop severe cutaneous inflammation, hyperkeratosis and acanthosis (*Blumberg et al., 2007*). Recently, work by Yang et al. showed that activated γδT-cells released IL-1F8, which had the ability to promote the proliferation of stem cells (*Blumberg et al., 2007*; *Yang et al., 2010*). As IL-1F6 and IL-1F8 signal through the IL-36 receptor and are able to induce the production of mitogenic factors, it is likely that abrogating IL-1R1 signaling alone may not be sufficient to block the epidermal hyperproliferation observed in the Casp8 cKO skin (*Towne et al., 2004*). Moreover, in a system as complex as the wound-healing program, removal of an individual component such as IL-1 signaling (*Kovalenko et al., 2009*) will impede the kinetics of the process rather than completely inhibit it (*Figure 1—figure supplement 1*). It is worth noting that γδT-cells have been found to stimulate hair neogenesis in wounds via Fgf9 (*Gay et al., 2013*). Interestingly, it is the dermal γδT-cells that secrete

Fgf9 whereas the epidermal pool of γδT-cells (DETCs) secrete IL-1F8, which does not possess hair-neogenesis-promoting properties (*Yang et al., 2010*). Consequently, it appears that there is heterogeneity among cutaneous γδT-cells that can secrete unique repertoires of cytokines with differential effects on the hair follicle.

Consistent with our observations in the skin, IL-1 and IL-7 can also work together to increase γδT-cell proliferation in the newborn thymus, where these cells compete with γδT-cells for survival (*Lynch and Shevach, 1992*). In the case of injury, this synergistic interaction is most probably a response to seal the wound quickly. Our findings suggest that in the presence of a wound-like state, the potentiation of γδT-cellproliferation may be a means to increase the pools of cutaneous epithelial stem cells. Our results are consistent with those of *Keyes et al., 2016*, which showed that optimal activation of γδT-cells is required for proliferation in hair follicle progenitors and efficient sealing of wounds (*Keyes et al., 2016*). Our results provide a mechanistic basis for their observation and it is interesting to note that the age-related decline that they found in wounding correlates with the documented changes in the IL-1 family of cytokines that occur with aging (*Ye et al., 2002*). Owing to the fact that HFSCs are highly activated in the Casp8 cKO skin, it is tempting to speculate that the hair regeneration cycle would be faster in this genetic background relative to the wild-type controls. Unfortunately, the truncated lifespan of the Casp8 cKO mouse (up to ~20 days) and the chronic inflammatory phenotype prevent an investigation of this possibility.

Given the inherent differences between IFE cells and HFSCs, it is perhaps not too surprising to find that they respond differently to signals from activated dermal fibroblasts and γδT-cells, respectively. Furthermore, the location of the proliferating inducing cell may impact which stem-cell pool it affects. For instance, in the backskin of WT mice, γδT-cells are found in close association with the HFSC niche (*Lee et al., 2009*). On the other hand, dermal fibroblasts are in close apposition to the IFE cells in the basal layer of the epidermis, which would be subjected to the highest concentration of pro-proliferation factors secreted by the activated fibroblasts.

There is an emerging paradigm that signals classically associated with immune cells are involved in regulating stem-cell homeostasis (*Chen et al., 2015*). The results of our experiments shed new insight into the complex nature of the repair process, which is a recurring theme in some pathological conditions such as cancer. We showed that the increased proliferation observed in the Casp8 cKO can be attained through reciprocal keratinocyte–fibroblast or keratinocyte–γδT-cell interactions. The keratinocyte–fibroblast-mediated proliferation is akin to the interaction between the tumor cell and stromal cells (*McAllister and Weinberg, 2010*). With regards to the keratinocytes–γδ T-cell interaction, it has recently been shown that the activation of γδT-cells promotes tumorigenesis (*Arwert et al., 2010*). Thus an understanding of the molecular mechanisms underlying the proliferation of stem cells in a wound microenvironment holds the promise of shedding light on the complex signaling pathways that mediate tumor initiation.

# Materials and methods

## Key resources table

| Designation | Source or reference | Identifiers | Additional information |
| --- | --- | --- | --- |
| Casp8 fl/fl mice (Casp8 cKO) | Jackson Laboratory | Stock No. 027002 | Animals were originally purchased from Jackson Laboratory but were bred for >5 generations at UCSD and later bred in the NCBS animal facility for >5 generations |
| IL1R KO mice | Jackson Laboratory | Stock No. 028398 | Animals were originally purchased from Jackson Laboratory (Stock No. 000664) but were bred for > 5 generations in the NCBS animal facility |
| Tcrd⁻/⁻ mice (γδTCR KO) | Jackson Laboratory | Stock No. 003448 | Animals were originally purchased from Jackson Laboratory (Stock No. 000664) but were bred for > 5 generations in the NCBS animal facility |
| Krt14-Cre | Jackson Laboratory | Stock No. 018964 | Animals were originally purchased from Jackson Laboratory (Stock No. 000664) but were bred for > 5 generations in the NCBS animal facility |

*Continued on next page*

*Continued*

| Designation | Source or reference | Identifiers | Additional information |
|---|---|---|---|
| C57Bl6/JNcbs | Jackson Laboratory | Stock No. 000664 | Animals were originally purchased from Jackson Laboratory (Stock No. 000664) but were bred for > 10 generations in the NCBS animal facility |
| anti-CD34 | Ebioscience | 11-0341-82 | (1:100) |
| anti-Ki67 | abcam | ab16667 | (1:100) |
| anti-alpha-smooth muscle actin (a-SMA) | Abcam | ab5694 | (1:100) |
| anti-gamma delta TCR | BD Bioscience | GL3 | (1:100) |
| anti-CD3 | e-biosciences | 14-0032-85 | (1:100) |
| anti-JAML | eBioscience | clone eBio4E10 | (1:100) |
| anti-IL-7 inhibitory antibody (Clone M25) | BioXCell | BE0048 | (1:100) |
| anti-IL-1 alpha | Invitrogen | 14-7011-81 | (1:100) |
| anti-Keratin 5 | Jamora Lab generated | | (1:500) |
| anti-Loricrin | Jamora Lab generated | | (1:500) |
| anti-Sox9 | Abcam | ab185230 | (1:100) |
| anti-Mac1 | Thermo-fisher scientific | MA1-10080 | (1:100) |
| anti-Gr1 | R&D Systems | MAB1037-500 | (1:100) |
| anti-TNFalpha | eBiosciences | 14–7321 | (1:100) |
| Alexa 488- or 555- secondaries | Molecular Probes | | (1:400) |
| DAPI | Molecular Probes | | (1:1000) |
| rhIL-1 alpha | R&D Systems | 200-LA-010 | |
| ELISA kit for IL-1 alpha | eBiosciences | 88501986 | |
| Click-iT EdU Imaging Kit | Thermo-fischer scientific | C10340 | |
| GraphPad | Prism | | For statistical analysis |

## Generation of caspase-8 conditional knockout mice (Casp8 cKO)

Epidermis-specific knockouts were obtained by crossing mice carrying the floxed caspase-8 allele (Casp8 fl/fl) (*Beisner et al., 2005*) to Krt14-Cre mice (*Vasioukhin et al., 1999*). IL1R$^{-/-}$ mice were purchased from Jackson Laboratory. γδTCR$^{-/-}$ mice have been described previously (*Jameson et al., 2002*). All animals were on a C57BL/6 background. Animal work carried out at UCSD was approved and adhered to the guidelines of the Institutional Animal Care and Use Committee. Animal work conducted in the NCBS/inStem Animal Care and Resource Center was approved by the inStem Institutional Animal Ethics Committee following norms specified by the Committee for the Purpose of Control and Supervision of Experiments on Animals, Government of India.

## Histology, immunohistochemistry and quantification

Mouse skin samples were processed and immunohistochemistry was performed as described previously (*Du et al., 2010*; *Lee et al., 2009*). Ki67 quantification was performed as described previously (*Arwert et al., 2010*).At least 12 frames per sample from four mice per genotype were used in counts.

## Quantitative real-time PCR

Epidermis and dermis from WT (n = 6), caspase-8 cKO (n = 9), caspase-8/IL1R dKO (n = 7) and caspase-8/γδT dKO (n = 5) P4 mice were separated with dispase treatment for 1 hr at 37°C and RNA was isolated using Trizol reagent (Invitrogen) according to manufacturer's instructions. cDNA was synthesized by reverse transcription using the iScript kit (Biorad), and real-time quantitative PCR analysis was performed using the Ssofast EvaGreen mix in a Biorad CFX96 system with the primers listed in the supplemental section. Experiments were carried out in triplicate with cDNA isolated from five different animals. Data are presented as the fold difference ± SEM.

## Wound-closure kinetics

8-week-old male IL1R$^{-/-}$ and WT control (C57Bl/6) mice were anesthesized by intraperitoneal injections with pentobarbital at 50 mg/kg. 5 mm punch biopsies were used to make full-thickness excisional wounds. Each day, images of the wounds were taken and analyzed using ImageJ software to measure wound area.

## In vivo labeling of proliferating cells with EdU

Postnatal day 6 (P6) mice were injected intraperitoneally with 50 µg/body weight EdU (5-ethynyl-2´-deoxyuridine, Invitrogen) dissolved in sterile PBS. Skins were collected 4 hr afterwards, embedded and frozen in OCT. 10 µm thick sections were fixed in 4% paraformaldehyde for 10 min, blocked in donkey serum for 1 hr and stained overnight with primary antibodies, followed by 30 min incubation with secondary antibodies. Proliferating HFSCs are recognized by co-staining CD34 and Ki67. However, CD34 is not expressed in skin at P6. Therefore, the location of HFSCs in the bulge region in P6 skin is marked by smooth muscle actin (SMA)-positive arrector pilli muscle, which joins the hair follicle at the bulge region where HFSCs reside (*Poblet et al., 2002*). γδT-cells were detected with anti-γδTCR (GL3, BD Biosciences).

## γδT-cell proliferation assays

Short-term cultures of WT and IL1R$^{-/-}$ γδT-cells were established and 30,000 cells were plated in 96-well plates coated with 0.1 µg/ml anti-CD3 and 10 µg/ml anti-JAML (eBioscience, clone eBio4E10) as described previously (*Witherden et al., 2010*). Cells were treated with conditioned media from WT and caspase 8 cKO keratinocytes collected in RPMI medium. γδT-cells were treated with KO CM or KO CM with 4 µg/ml anti-IL-7 inhibitory antibody (R&D Systems) or IL1R$^{-/-}$ γδT-cells were treated with KO CM. γδT-cells were incubated with WT CM, WT CM treated with 300 pg/ml rhIL-1α, or media supplemented with rhIL-1α and 2 ng/ml rhIL-7. Cell numbers were assessed by cell counting using trypan blue exclusion.

## Epidermal stem-cell proliferation assays

Unipotent interfollicular epidermal (IFE) stem cells were isolated as described previously *Lee et al., 2009*. Hair follicle stem cells were generated by DrsEve Kandyba and Krzysztof Kobielak at USC and prepared as described previously (*Blanpain et al., 2004*). Dermal fibroblasts were isolated from 6-week-old C57bl/6 WT and IL1R$^{-/-}$ mice. The hairs were removed by shaving and treatment with a hair removal agent (Nair, Church and Dwight Co., Princeton, NJ, USA). Epidermis and dermis were separated after overnight incubation with trypsin solution at 4°C. Dermal portions were then incubated in collagenase IV for 1 hr at 37°C with shaking, and collagenase was neutralized with 2 mM EDTA. Cell suspensions were filtered through a 70 µm cell strainer and pelleted by centrifugation at 400 x g for 10 min at 25°C. The cells were then plated in DMEM with 10% FBS, 1% penicillin/streptomycin and 2 mM L-glutamine. WT and IL1R$^{-/-}$ dermal fibroblasts were treated with WT and caspase 8 KO CM in the presence of rhIL-7, rhIL-1α or anti-IL-7 inhibitory antibody for 24 hr. 5000 IFE and HF stem cells were plated in 48-well plates and incubated with CM from treated dermal fibroblasts for 4 days. Cell counts were assessed by trypan blue exclusion.

## Activated γδT-cell assay in vivo

P4 WT, caspase 8 cKO and caspase 8/IL1R dKO mice were injected with 0.2 mg of Brefeldin A (BFA, Sigma-Aldrich, St. Louis, MO, USA) subcutaneously and samples collected after 5 hr. Skins were frozen with OCT compound (Tissue-Tek) and 10 µm sections were made. Tissues were fixed for 10 min in 4% PFA, blocked (in 2.5% goat serum, 2.5% donkey serum, 1% BSA, and 0.3% triton-X in PBS) and incubated for 1 hr at 25°C with antibodies against γδTCR-FITC (GL3, BD Biosciences) and TNFα-PE (eBiosciences, San Diego, CA). Images were obtained as described previously (*Du et al., 2010*).

## Oligos used for qPCR

TCRVγ3-F: GCAGCTGGAGCAAACTGAAT
TCRVγ3-R: GTTTTTGCCGGTACCAATGT
FGF7-F: GTGAGAAGACTGTTCTGTCGC
FGF7-R: CCACGGTCCTGATTTCCATGA

FGF10-F: GTGTCCTGGAGATAACATCAGTG
FGF10-R: AGCCATAGAGTTTCCCCTTCTT
TNF⟨-F: CTGTGAAGGGAATGGGTGTT
TNF⟨-R: GGTCACTGTCCCAGCATCTT
IL1F6-F: CACGTACATGGGAGTGCAAA
IL1F6-R: GCAGCTCCCTTTAGAGCAGA
IL1F8-F: GGTATGGGTCCTGACTGGAA
IL1F8-R: CCTCCATCTCAACACAGCAG
IL7-F: TGGAATTCCTCCACTGATCC
IL7-R: TGGTTCATTATTCGGGCAAT

## ELISA

8-week-old WT (C57BL6) and IL1R$^{-/-}$ mice were used for wounding and conditioning of epidermal stem cell media lacking serum. 5 mm punch biopsies was used to make full-thickness excisional wounds. Wound samples were collected from an 8-mm biopsy surrounding the wound. 200 μl of epidermal stem cell media lacking serum was added to the wound samples and incubated at 37°C for 1 hr. The medium was replaced with another 200 μl of epidermal stem cell media lacking serum and incubated at 37°C for 16 hr, and this was used as conditioned media. Conditioned media was collected and stored at −80°C after snap-freezing in liquid $N_2$. IL1α ELISA was performed using the mouse IL1-alpha ELISA kit (eBiosciences, San Diego, CA, USA) according to the manufacturer's protocol. Conditioned media from wounds were diluted 1:5 and 1:15 to detect IL1α secretion. Absorbance was detected using the Infinite 200 Pro (Tecan, Männedorf, Switzerland) microplate reader. Total IL1α secretion was estimated by comparing the absorbance values of samples to the standard curve.

## Statistical analysis

All in vivo experiments were done on three mice per genotype and samples in in vitro assays were run in triplicate. Results were generated by average ± SEM from three independent experiments. For comparison of means between two groups, Student $t$ test was performed. For multigroup comparisons, ANOVA test with a Bonferroni's multiple comparison correction was used. All p-values < 0.05 were considered significant.

Further details regarding reagents are provided in the Research Resource Identifiers document.

## Acknowledgements

The authors would like to thank Dr. David Stachura for assistance with FACS analysis and Dr. Hans Dietmar-Beer and the Jamora lab for helpful discussions. PL was supported by a predoctoral fellowship from the NIH (5F31AR056593), RG is supported by a Department of Biotechnology (DBT) (Government of India) Research Associate Fellowship, IR is supported by an ICMR Senior Research Fellowship, and SG is supported by a Wellcome Trust–DBT Alliance Early Career Fellowship. This work was supported by a Hellman Faculty Fellowship and by grants from the NIH/NIAMS (Grant 5R01AR053185-03) and the American Cancer Society (Grant 115457-RSG-08-164-01-DDC) to CJ, and by grants from the NIH to WH (R01AI036964 and R01AI064811) and AM (T32 AI007244). Animal work was partially supported by the National Mouse Research Resource (NaMoR) grant (BT/PR5981/MED/31/181/2012;2013–2016) from the DBT.

## Additional information

### Funding

| Funder | Grant reference number | Author |
|---|---|---|
| National Institutes of Health | 5F31AR056593 | Pedro Lee |
| Government of India | Department of Biotechnology Research Associate fellowship | Rupali Gund |

| Indian Council of Medical Research | Senior Research Fellowship | Isha Rana |
|---|---|---|
| Wellcome | DBT Alliance Early Career Fellowship | Subhasri Ghosh |
| National Institutes of Health | T32 AI007244 | Amanda MacLeod |
| National Institutes of Health | R01AI036964 | Wendy L Havran |
| National Institutes of Health | R01AI064811 | Wendy L. Havran |
| National Institute of Arthritis and Musculoskeletal and Skin Diseases | 5R01AR053185-03 | Colin Jamora |
| American Cancer Society | 115457-RSG-08-164-01-DDC | Colin Jamora |
| Department of Biotechnology, Ministry of Science and Technology | | Colin Jamora |
| Hellman Foundation | Hellman Faculty Fellowship | Colin Jamora |

The funders had no role in study design, data collection and interpretation, or the decision to submit the work for publication.

## Author contributions

Pedro Lee, Rupali Gund, Conceptualization, Data curation, Formal analysis, Investigation, Methodology; Abhik Dutta, Subhasri Ghosh, Conceptualization, Formal analysis, Investigation, Methodology; Neha Pincha, Isha Rana, Deborah Witherden, Eve Kandyba, Amanda MacLeod, Formal analysis, Investigation, Methodology; Krzysztof Kobielak, Funding acquisition, Investigation, Methodology; Wendy L Havran, Conceptualization, Supervision, Funding acquisition, Investigation, Methodology; Colin Jamora, Conceptualization, Data curation, Formal analysis, Supervision, Funding acquisition, Investigation, Methodology, Writing—original draft, Writing—review and editing

## Author ORCIDs

Colin Jamora http://orcid.org/0000-0003-2127-2972

## Ethics

Animal experimentation: Experimental work was approved by the Institutional Biosafety Committee of the Institute of Stem Cell Biology and Regenerative Medicine for studies on "Mechanisms regulating wound healing in the skin and the diseases that arise when this process is perturbed" (Certificate No. inStem/G-141(3)/2012). The project involving animal work was approved by the Institutional Animal Ethics Committee of the Institute of Stem Cell Biology and Regenerative Medicine for studies on the "Regulation of Skin and Hair Development, Regeneration and Repair" (Certificate No. INS-IAE-2016/17(ME)). Animal work carried out at UCSD was approved and adhered to the guidelines of Institutional Animal Care and Use Committee. Animal work conducted in the NCBS/inStem Animal Care and Resource Center was approved by the inStem Institutional Animal Ethics Committee following norms specified by the Committee for the Purpose of Control and Supervision of Experiments on Animals, Govt. of India.

## Decision letter and Author response

Decision letter https://doi.org/10.7554/eLife.28875.032
Author response https://doi.org/10.7554/eLife.28875.033

# Additional files

## Supplementary files

• Transparent reporting form
DOI: https://doi.org/10.7554/eLife.28875.030

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
