## [Decision Letter]

Thank you for submitting your article "Stimulation of hair follicle stem cell proliferation through an IL-1 dependent activation of γδT-cells" for consideration by *eLife*. Your article has been reviewed by two peer reviewers, and the evaluation has been overseen by Fiona Watt as the Senior and Reviewing Editor. The reviewers have opted to remain anonymous.

The reviewers have discussed the reviews with one another and the Reviewing Editor has drafted this decision to help you prepare a revised submission.

Summary:

Recent research has shown that interactions between immune cells, epithelial cells and stromal cells play a crucial role in tissue repair and regeneration. The present work is of interest because it provides more experimental information about those interactions. While γδT cells have been shown previously to be involved in wound-induced hair neogenesis, the present paper shows activation of γδT cells in a different wound setting. Their results lead the authors to propose a new model whereby inflammation can differentially affect different lineages of epidermal stem cells during wound-induced regeneration.

Essential revisions:

1) It has been noticed that IL-1 plays an important role in stimulating regeneration following wounding. The authors try to demonstrate that this is via activation of γδT cells. To eliminate the possibility IL-1 exerts its effect without γδT cells the authors should inject IL-1α and IL-7 into γδT cells KO mice to see if these two cytokines can rescue the delayed wound healing process in the absence of γδT cells.

2) The hair follicle status in Figure 1 and Figure 2 is quite different. The authors use telogen hair follicles as a model in Figure 1 but anagen hair follicles in Figure 2. It is known that the hair cycle involves changes in epidermal cell proliferation. Please clarify the age or status of hair regeneration cycle of the transgenic mice used in all experiments. If the cycling status of hair follicle is not the same in different experiments, the results are uninterpretable.

3) Both the caspase mutation and tissue culture studies are fairly contrived environments and so the authors should make the effort to examine wounds in wildtype mice and directly test whether they have uncovered real in vivo mechanisms.

4) The acute in vivo wound healing response is more about cell migration than cell proliferation and at points in the paper cell migration should be investigated; however, the authors focus entirely on proliferation. This is something that could be addressed in culture.

Other points:

1) Previous work has shown that wounded skin can produce CCL2 through ASK-1 to attract macrophages, activated by IL-1β, to induce hair regrowth (Osaka et al., 2007). Here, the authors showed that γδT cells play major roles in wound induced hair stem cell activation upon the stimulation of IL-1α. Can the authors reconcile this difference, and discuss whether macrophages, γδT cells or both, are the key factors? Also, can authors clarify which cells are at higher hierarchical role in this process?

2) A previous study demonstrated that γδT cells can secrete FGF-9 to stimulate hair regeneration or activate hair follicle stem cells in wounds (Gay et al., 2013). In this study, the authors propose that IL1F8 or IL-36β released from γδT cells are the major signals involved in stem cell activation. Can the authors reconcile the different observations?

3) The authors need to appropriate statistical evaluation of their results throughout the paper.

4) This was not a particularly easy paper to read because of the long chunks of text without breaks. In places it is difficult to follow the logic of experiments, and to understand which of these have already been published versus those that are part of the current study.

---

## [Author Response]

Essential revisions:1) It has been noticed that IL-1 plays an important role in stimulating regeneration following wounding. The authors try to demonstrate that this is via activation of γδT cells. To eliminate the possibility IL-1 exerts its effect without γδT cells the authors should inject IL-1α and IL-7 into γδT cells KO mice to see if these two cytokines can rescue the delayed wound healing process in the absence of γδT cells.

This is an important point that we have addressed in a number of ways. The suggested experiment to inject IL-1α and IL-7 into the γδT cellKO mouse is ideal, however we found that the administration of a bolus of IL-1α into the skin caused a massive inflammatory response that led to considerable tissue damage in a relatively short time period. We likewise found a similar effect upon multiple administrations of lower doses of IL-1α that preclude any investigation on rescuing the delayed wound healing process in the absence of γδT-cells. In retrospect, this was not surprising given an earlier report that transgenic overexpression of IL-1α in epidermal keratinocytes caused a spontaneous and robust inflammatory response that led to “hair loss, scaling, and focal inflammatory skin lesions” (Groves et al., 1995). Likewise, previous reports of the intradermal injection of IL-1α showed a very rapid and substantial proinflammatory phenotype (Ramparts and William, 1988; Camp et al., 1988). Importantly, it has also been reported that IL-1 levels naturally fluctuate during the wound-healing program (Kondo and Ohshima, 1996). For instance, the IL-1 α levels reached a peak at 6 hours post wounding, then decreases until there is another spike in IL-1 levels at 72 hours. Consequently, injection studies of IL-1α would not recapitulate the dynamic physiological variations in cytokine levels that are characteristic of a normal wound healing response in vivo and would probably be a model more relevant to a pathological condition.

In order to address the reviewers’ concern, we have included a number of new experiments in the revised manuscript to further support the model that IL-1α’s effect on hair follicle stem cell proliferation is primarily through γδT-cells. In the original version of the manuscript, we demonstrated that the absence of γδT-cells abrogated the effect of wound-stimulated hair follicle stem cell proliferation. We now provide evidence that the IL-1α and IL-7 levels remain the same as in the wild type wounded skin despite lacking functional γδT-cells. This substantiates our conclusion that despite the physiological levels of these cytokines being present, the impairment of HFSCs proliferation is still manifested in the absence of γδT-cells.

These are shown in the following:

Figure 2—figure supplement 5: Immunofluorescence of unpermeabilized skin sections to detect extracellular IL-1α in wild type, caspase-8 cKO, and caspase8cKO/γδT-cell KO;

Figure 2—figure supplement 6: ELISA measurement of conditioned media from unwounded, wounded, and γδT-cell KO wounded skins;

Figure 2—figure supplement 7: Immunofluorescence of unpermeabilized skin sections to detect extracellular IL-1α in unwounded, wounded, and γδT-cell KO wounded skins;

Figure 3—figure supplement 4 and Figure 3—figure supplement 5: Assay of the levels of IL-7 via qPCR in wild type wounded, and γδT-cell KO wounded skins;

Altogether these point to the fact that despite the normal levels of IL-1α and IL-7 in the wound, active γδT-cells are necessary to elicit the early stimulation of hair follicle stem cell proliferation.

2) The hair follicle status in Figure 1 and Figure 2 is quite different. The authors use telogen hair follicles as a model in Figure 1 but anagen hair follicles in Figure 2. It is known that the hair cycle involves changes in epidermal cell proliferation. Please clarify the age or status of hair regeneration cycle of the transgenic mice used in all experiments. If the cycling status of hair follicle is not the same in different experiments, the results are uninterpretable.

We apologize for the confusion caused by not clearly delineating the age of the mice and the status of the hair cycle in the different models of wound healing. As suggested, we have detailed this information for the mice used in all experiments. Briefly, due to the strong phenotype of the genetic model of wound healing (epidermal caspase-8 conditional knockout) the animal rarely survives pass postnatal day 20. As we previously reported (Lee et al., 2009), the wound healing like phenotypes start manifesting at postnatal day 6 when the hair follicle is in anagen. For this reason, the hair follicles are in the anagen phase in Figure 2. In Figure 1, the excisional wounds are done on adult (8 week old) mice as standard protocol. As the reviewer keenly observed, these hair follicles are in telogen. Of particular note is the fact that the effect of the IL-1α/γδTcell signaling module on hair follicle stem cell proliferation is independent of the hair phase. Nevertheless, to address the reviewer’s concern, we have performed excisional wounds on adult animals in which the hair follicle was in anagen phase and the results are exactly the same as we reported in the original submission. Due to space issues (and the fact that all the experiments that were performed on adult mice with anagen follicles are confirmatory of data presented in the manuscript) we have noted these results as “data not shown” in the Discussion section.

3) Both the caspase mutation and tissue culture studies are fairly contrived environments and so the authors should make the effort to examine wounds in wildtype mice and directly test whether they have uncovered real in vivo mechanisms.

The reviewer makes a cogent point and we have repeated all of the caspase-8 conditional knockout experiments with excisional wounds. We were pleased to observe that all of the signaling pathways that we have elucidated using this genetic model of wound healing is indeed a recapitulation of the signaling pathways that are operational following an excisional wound on the mouse skin. The following experiments have been replicated in the excisional wound model (that were shown in the caspase-8 cKO genetic wound healing model in the original submission).

4) The acute in vivo wound healing response is more about cell migration than cell proliferation and at points in the paper cell migration should be investigated; however, the authors focus entirely on proliferation. This is something that could be addressed in culture.

The reviewer is absolutely correct in highlighting the importance of cell migration in the wound healing response and we thank him/her for bringing this point to the fore. Given that in vivo hair follicle stem cell migration experiments in a live animal is beyond our technical capabilities, we have recapitulated this process in vitro. The reconstitution of migration in vitro also facilitates the unambiguous separation of the effect of proliferation from migration on the number of hair follicle stem cells that are outside the bulge region of the follicle. The sum of these experiments is a clear demonstration that cell migration is independent of IL-1 signaling and its downstream targets. Briefly, as we show in Figure 1—figure supplement 7, we are able to induce hair follicle stem cell chemotaxis in the classic transwell assay using conditioned media from the epidermis of the genetic model of wound healing (caspase-8 conditional KO). Likewise, conditioned media from the caspase-8cKO/ γδT-cells KO was also able to induce stem cell migration, thereby showing the dispensability of γδT-cells in migration. However, using recombinant IL-1α as a chemoattractant in a transwell assay was not sufficient to induce directed migration of hair follicle stem cells. Importantly, the conditioned media from the caspase-8 cKO epidermis was still able to induce migration of hair follicle stem cells lacking the IL-1 receptor. Altogether this provides evidence that IL-1α is neither necessary nor sufficient to induce hair follicle stem cell migration. We are actively searching for the component(s) in the caspase-8 conditioned media that is responsible for the migration, but this falls beyond the purview of the current manuscript.

Other points:1) Previous work has shown that wounded skin can produce CCL2 through ASK-1 to attract macrophages, activated by IL-1β, to induce hair regrowth (Osaka et al., 2007). Here, the authors showed that γδT cells play major roles in wound induced hair stem cell activation upon the stimulation of IL-1α. Can the authors reconcile this difference, and discuss whether macrophages, γδT cells or both, are the key factors? Also, can authors clarify which cells are at higher hierarchical role in this process?

The reviewer highlights an interesting point, which we now address in the Discussion section of the manuscript. In brief, there is cumulative contribution from both γδTcells and macrophages, as pointed out in Osaka et al. 2007. The model that emerges is that IL-1α is immediately released from their reservoirs upon keratinocyte injury and activates γδTcells (along with IL-7). This early response may be subsequently maintained (and amplified) by macrophages that are later recruited through the production of CCL-2 from stressed keratinocytes. These macrophages can then secrete IL-1β, which is a more potent cytokine than IL-1α, and thereby accentuate and maintain the elevated proliferative capacity of the hair follicle stem cells. As such, the hierarchical model that emerges is a temporal one: IL-1α is an early and local player in wound healing that leads to the activation of γδT-cells to stimulate HFSC proliferation, which is subsequently promoted by the infiltration of IL-1β secreting macrophages.

2) A previous study demonstrated that γδT cells can secrete FGF-9 to stimulate hair regeneration or activate hair follicle stem cells in wounds (Gay et al., 2013). In this study, the authors propose that IL1F8 or IL-36β released from γδT cells are the major signals involved in stem cell activation. Can the authors reconcile the different observations?

The reviewer raises an important issue, which we now address in the Discussion section of the manuscript. Based on our data, the model that arises is that the γδTcells associated with the epidermis and hair follicle (the so-called dendritic epidermal T-cells [DETCs]) secrete IL1F8 or IL-36β based on the literature (Blumberg et al., 2007; Yang et al., 2010). The Gay et al., manuscript demonstrates that it is the *dermal* γδT-cells that are the prime source of *FGF9* in wounded skin. Moreover, they also report that DETCs do not make *FGF9*. Thus the differential source of cytokines may explain the different types of cytokines that are released.

3) The authors need to appropriate statistical evaluation of their results throughout the paper.

All the results are now presented with appropriate statistical evaluation.

4) This was not a particularly easy paper to read because of the long chunks of text without breaks. In places it is difficult to follow the logic of experiments, and to understand which of these have already been published versus those that are part of the current study.

We have revised major portions of the manuscript to render the text more easily readable in which the rationale of each experiment is clearly delineated. Moreover we have paid special attention to described reported results vs. those that are part of the current study.